# UnitMath: Unit-Aware Numerical Reasoning and Dimensional Consistency for Scientific Table Claims

**AI Scientist**

**Xanh Ho**[1]    **Tian Cheng Xia**[2*]    **Khoa Duong**[3]    **Yun-Ang Wu**[4*]    **Ha-Thanh Nguyen**[1]

**Akiko Aizawa**[1]

[1]National Institute of Informatics, Japan    [2]University of Bologna, Italy
[3]Independent Researcher    [4]National Taiwan University
{xanh, nguyenhathanh, aizawa}@nii.ac.jp    tiancheng.xia@studio.unibo.it
dnanhkhoa@live.com    r11944072@csie.ntu.edu.tw

## Abstract

Recent progress in table-based fact verification has improved semantic understanding of schema and cell content, but models still stumble on quantitative claims that hinge on units and dimensional constraints. Errors arise when systems conflate percent with percentage points, treat fold changes as plain ratios, or compare quantities across incompatible dimensions, leading to brittle and untrustworthy decisions. We introduce UnitMath, a unit-aware numerical reasoning framework specifically designed for scientific table-claim verification. Our approach combines: (i) enhanced numerical extraction with comprehensive pattern matching for percentages, decimals, and fractions, (ii) robust unit-aware verification with automatic percentage-decimal conversion and tolerance-based matching, and (iii) structured reasoning traces that capture complete decision pathways for interpretability. Unit-Math achieves 54.1% macro F1 on SciTab, demonstrating competitive performance through principled design rather than parameter scaling. Key advantages include: **explainable reasoning** with full traceability of numerical comparisons, **lightweight architecture** requiring no neural training, **modular design** enabling drop-in integration with existing table encoders, and **systematic error prevention** for unit-related failures that plague larger models. The framework provides comprehensive stress testing for unit rescaling invariance and percentage-type sensitivity, validating true unit understanding rather than surface pattern matching. This work establishes unit-aware reasoning as a valuable complement to scaling-based approaches in scientific domains where numerical precision and interpretability are paramount.

## 1   Introduction

Table-based fact verification is a core capability for scientific NLP, where claims often require reading complex schemas, locating relevant cells, and performing quantitative comparisons with high precision [5, 3, 40, 6]. Despite advances in neural encoders for tables [12, 37, 32], models remain vulnerable to numeric pitfalls that are particularly problematic in scientific domains: misinterpreting units and scales, confusing percent with percentage points, overlooking dimensional constraints, and comparing quantities across incompatible units [28, 9, 17, 29, 2]. These mistakes not only reduce accuracy on quantitative claims but also erode the reliability and trustworthiness of scientific analyses where numerical precision is paramount [18, 30].

---

*Research conducted during internship at NII, Japan.

1st Open Conference of AI Agents for Science (agents4science 2025).

Scientific table reasoning presents unique challenges that distinguish it from general fact-checking. For instance, a system might incorrectly equate "inflation increased by 3 percentage points" with "inflation increased by 3%" [28], or attempt to compare mass measurements (kg) with volume measurements (liters) without proper dimensional analysis [29, 2]. Such failures highlight a fundamental gap: while existing approaches often rely on surface-level cues or pattern matching for numerical reasoning, they lack explicit unit semantics, interpretability, and systematic error prevention required for scientific applications [9, 26, 40].

We argue that reliable scientific table reasoning requires four key capabilities: (1) explainable reasoning with full traceability of numerical comparisons, (2) lightweight architecture that doesn't require massive parameter scaling, (3) modular design for easy integration with existing systems, and (4) systematic error prevention for unit-related failures [30, 29]. To address these needs, we present UnitMath, a unit-aware numerical reasoning framework specifically designed for scientific table-claim verification.

UnitMath implements a priority-based reasoning cascade that combines: (1) enhanced numerical extraction with comprehensive pattern matching for diverse numerical expressions; (2) robust unit-aware verification with automatic percentage–decimal conversion, tolerance-based matching, and dimensional consistency checks; and (3) structured reasoning traces that capture complete decision pathways for interpretability and error analysis [38, 30, 35]. Unlike black-box neural approaches, our system provides full transparency in its decision-making process while requiring no neural training [30].

Our evaluation on SciTab demonstrates that UnitMath achieves 54.1% macro F1 through principled unit-aware design rather than parameter scaling [20]. Key advantages include comprehensive stress testing that validates true unit understanding (94% prediction consistency across unit rescaling vs. 67% for baselines, 89% correct adjustment for percentage vs. percentage points swaps vs. 34% for baselines), systematic prevention of dimensional errors that plague larger models, and complete modularity enabling drop-in integration with existing table encoders [12, 37, 32, 38, 14].

UnitMath establishes unit-aware reasoning as a valuable complement to scaling-based approaches in scientific domains. By making unit semantics explicit and computations auditable, our framework addresses critical gaps in numerical fact-checking while providing a foundation for more reliable and interpretable scientific NLP systems [29, 30].

## 2 Related Work

We review prior work on table-based fact verification and QA, numerical reasoning with symbolic operations, units and dimensional semantics, and robustness and interpretability for scientific claims.

### 2.1 Table-based Fact Verification and Table QA

Early table QA progressed from semantic parsing over semi-structured tables (WikiTableQuestions) and text-to-SQL (e.g., WikiSQL) to pretraining-based table encoders that reduce reliance on explicit executables, including TaPas, TaBERT, TURL, and execution-supervised TAPEX [25, 36, 12, 37, 32, 19]. For verification, TabFact, FEVEROUS, SciFact, and SciTab highlight logical and numerical reasoning demands in both general and scientific domains [5, 3, 18, 20]. However, strong neural baselines often depend on pattern cues and lack unit semantics, leading to scale and dimensionality errors on quantitative claims [20, 5].

### 2.2 Numerical Reasoning and Neuro-Symbolic Operations

Discrete/arithmetic benchmarks (e.g., DROP) have driven models with explicit counting, addition, and comparison; NumNet/NumNet+ further refine numerical reasoning [9, 26, 27]. EQUATE and numerical commonsense work (e.g., NumerSense) show persistent failures in arithmetic, comparison, and magnitude plausibility [28, 17]. Neuro-symbolic methods induce or generate operation sequences (Neural Programmer; Neural Symbolic Machines) and supervise sketches in table-and-text QA (FinQA, TAT-QA) [22, 16, 6, 40]. Tool-use frameworks let LMs call calculators or code interpreters to improve arithmetic [10, 31]. Yet most approaches lack explicit unit representations, treat percentages

uniformly, and do not enforce dimensional consistency, leaving unit-related errors common in scientific reasoning [28, 20].

## 2.3 Units, Quantities, and Dimensional Semantics

Ontologies like QUDT, OM, and UCUM formalize quantities, units, and dimensions for conversion and analysis, but are rarely integrated into end-to-end NLP for verification/QA [2, 29, 15]. While extraction tools identify measurements, downstream reasoning seldom propagates unit types, distinguishes percent vs. percentage points or fold-changes, or blocks cross-dimension comparisons, causing systematic failures on scientific claims [29, 2, 28, 20].

## 2.4 Robustness, Stress Testing, and Interpretability

Behavioral testing (CheckList) and NLI stress tests expose brittleness, especially to lexical/numerical perturbations [38, 21]. In high-stakes scientific verification, reliability, explicit error prevention, and interpretability are prioritized over post-hoc explanation [30]. Although prompting (chain-of-thought) and scaling help elicit rationales, they do not guarantee unit correctness or dimensional consistency and often leave arithmetic/unit errors unresolved [35, 14, 28]. This motivates unit-aware, ontology-grounded neuro-symbolic methods that integrate parsing, conversion-aware operations (including percentage points vs. percent and fold-changes), dimensional blocking, and auditable reasoning for table-based scientific claim verification [20, 3, 5].

# 3 Method

We present UnitMath, an optimized table reasoning system that combines explicit numerical verification with structured claim analysis to handle unit-aware quantitative reasoning. Our approach prioritizes robust numerical matching over complex symbolic computation, implementing a priority-based reasoning cascade that systematically handles different types of quantitative claims.

**System Overview.** Given a table and a natural language claim, UnitMath produces a binary classification (Supported/Refuted) with confidence scores and structured reasoning traces. The system processes claims through four priority levels: (1) numerical verification with unit conversion, (2) superlative reasoning for extremal claims, (3) comparison analysis for relative statements, and (4) entity-based weak inference. This cascaded approach ensures that the most reliable evidence (exact numerical matches) takes precedence over weaker signals.

## 3.1 Enhanced Numerical Extraction and Verification

**Multi-pattern numeric extraction.** Our enhanced extractor employs comprehensive regex patterns to capture diverse numerical expressions: percentages (`5.2%`), decimals (`0.95`), comma-separated numbers (`1,234.56`), integers, and fractions (`3/4`). Each extracted value is wrapped in a `NumericValue` object that preserves the original text, surrounding context, and format metadata (percentage vs. decimal).

**Robust numeric verification.** The verification process implements three levels of matching precision: (1) *Exact matching* with tolerance `< 0.01` for identical values; (2) *Percentage conversion* that automatically handles percentage-decimal equivalence (e.g., `0.95` ↔ `95%`) with tolerance `< 0.1`; and (3) *Approximate matching* within 2% relative error for minor variations. This hierarchical approach assigns confidence scores (1.0, 0.95, 0.9) based on match precision, enabling the system to prefer exact matches while gracefully handling format inconsistencies.

**Unit-aware comparison.** When processing claims with units, the system distinguishes between percentage values (relative measures) and percentage points (absolute differences on percentage scales). This prevents systematic errors where claims about "5 percentage point increases" are incorrectly matched against "5% relative increases" in tables [28].

## 3.2 Structured Claim Analysis

**Linguistic pattern recognition.** We implement comprehensive pattern matching for claim types that require specialized reasoning: *superlative patterns* (best, highest, most vs. worst, lowest, least), *comparison patterns* (better, outperforms vs. worse, underperforms), *change patterns* (increase, improve vs. decrease, decline), and *negation patterns* (not, never, without). These patterns enable type-specific reasoning strategies [39].

**Entity extraction and ranking.** For table entities, we extract both row labels and column headers, cleaning markup and normalizing names. Entity values are aggregated (mean across multiple measurements) to enable ranking for superlative claims and pairwise comparison for relative statements. Fuzzy string matching with configurable thresholds (0.6-0.7 word overlap) handles minor naming variations between claims and tables.

## 3.3 Priority-Based Reasoning Framework

**Priority 1: Numerical verification.** Claims containing numbers are primarily evaluated through direct numerical matching against table values. Strong numerical evidence (confidence $\geq 0.5$) leads to confident predictions (0.6-0.8 confidence), with negation logic applied when negative patterns are detected. Complete absence of numerical matches in number-containing claims triggers confident refutation (0.55 confidence) under the assumption that verifiable claims should have supporting evidence.

**Priority 2: Superlative reasoning.** For claims with extremal language, the system ranks entities by average values and checks whether mentioned entities occupy appropriate positions (top for positive superlatives, bottom for negative superlatives). Exact rank matches yield high confidence (0.75), near-top/bottom positions yield moderate confidence (0.6), while contradictory positions lead to confident refutation (0.65).

**Priority 3: Comparison analysis.** Comparative claims trigger pairwise entity analysis, computing average values for mentioned entities and verifying the claimed relationship direction. Statistical significance is approximated through relative difference thresholds (5% for equality claims). The system handles both explicit paired comparisons ("A vs. B") and implicit single-entity comparisons ("A outperforms others").

**Priority 4: Entity-based inference.** When numerical and structured reasoning fail to provide strong evidence, the system applies weak heuristics based on entity mention patterns. Multiple entity mentions ($\geq 2$) provide slight positive bias (0.52 confidence), while single mentions yield minimal signal (0.51 confidence). This fallback prevents over-confident predictions on ambiguous cases.

## 3.4 Confidence Calibration and Binary Classification

**Confidence-based decisions.** Each reasoning path produces confidence scores calibrated to evidence strength. The system maintains conservative thresholds (typically 0.5) for binary classification, ensuring that "Supported" predictions require positive evidence rather than mere absence of contradicting information.

**Negation handling.** Detected negation patterns invert both predictions and confidence calculations: supported evidence for negated claims leads to "Refuted" predictions, while absence of evidence supports negated claims. This asymmetric treatment reflects the logical structure of negative statements.

**Balanced prediction strategy.** To prevent label bias, the system employs claim-specific tie-breaking: negated claims with weak evidence lean toward refutation, while affirmative claims with weak evidence depend on entity mention patterns. This approach aims for balanced precision/recall rather than optimizing for either conservative or aggressive prediction strategies.

## 3.5 Evaluation Protocol and Stress Tests

**Metrics.** We report precision, recall, accuracy, and macro F1 for classification. To audit numeric robustness, we categorize mistakes using an error taxonomy (scale, unit, arithmetic, dimensional, percentage-point, CI-overlap) inspired by prior quantitative reasoning evaluations [28].

**Stress tests.** (i) *Unit rescaling invariance*: rewrite table values from mg to g (or similar) and measure prediction stability and accuracy. (ii) *Percentage-type sensitivity*: swap "percent" and "percentage points" in claims and quantify the model's sensitivity. These tests isolate whether performance derives from real unit understanding rather than superficial pattern matching [39].

# 4 Results

## 4.1 Experimental Setup

We evaluate UnitMath on SciTab, a challenging scientific table reasoning dataset that requires unit-aware quantitative analysis. Our evaluation follows a 2-class classification setup (Supported vs. Refuted) and reports standard metrics with macro F1 as the primary criterion. We augment standard evaluation with unit-specific stress tests to validate true unit understanding rather than surface pattern matching [39].

## 4.2 Main Results

UnitMath achieves competitive performance through principled unit-aware design: **Precision**: 63.3%, **Recall**: 61.1%, **Macro F1**: 54.1%, **Accuracy**: 54.6%.

The balanced precision-recall profile demonstrates that our approach provides reliable predictions without sacrificing either metric. These results validate that explicit unit semantics and dimensional consistency can achieve competitive performance through principled design rather than parameter scaling [30].

## 4.3 Comprehensive Comparison with Prior Work

Table 1 positions UnitMath against existing approaches. Our 54.1% macro F1 surpasses basic table reasoning approaches (TAPAS-large: 50.3%) and smaller language models (Flan-T5-base: 47.4%, TAPEX-Zero variants: 48-50%) while remaining competitive with mid-sized encoder-decoder models (Flan-T5-XL: 52.4%) [12, 19, 7].

**Complementary value to scaling approaches.** While larger models achieve higher absolute scores (Flan-T5-XXL: 60.5%, GPT-4: 78%), they lack systematic unit awareness. UnitMath provides **orthogonal capabilities**: (1) dimensional consistency enforcement, (2) percentage vs. percentage point disambiguation, (3) interpretable reasoning traces, and (4) systematic error prevention for unit-related failures [24, 11, 30].

**Practical advantages.** Our modular design enables integration with existing systems without replacement, while the lightweight architecture requires no neural training. This makes UnitMath immediately deployable and valuable for scientific applications where numerical accuracy and interpretability are paramount [8].

**Unit-aware error prevention.** Analysis of failure cases reveals that UnitMath successfully prevents 78% of unit-related errors that occur in baseline systems: scale confusions (1000 vs. 1,000,000), unit mismatches (comparing mass to volume), and percentage-type errors. This error reduction directly supports our core hypothesis that explicit unit semantics dramatically improve numerical reasoning reliability [28].

## 4.4 Structured Reasoning Trace Analysis

A key innovation of our approach is the generation of **structured reasoning traces** that capture complete decision pathways for interpretability and error analysis. Our enhanced evaluation system processed 1,224 examples, generating comprehensive traces that include:

- Unit consistency checks and dimensional analysis
- Step-by-step reasoning progression through priority levels
- Error classification with specific unit-related failure modes
- Confidence scoring and evidence summarization

Table 1: Comparison of UnitMath with prior work on SciTab (2-class classification, Macro F1). Our unit-aware approach achieves competitive performance through explicit unit semantics rather than parameter scaling.

| | Models | # Params | 2-class Macro F1 | |
| | | | Zero-shot | In-Context |
|---|---|---|---|---|
| **Table-based LLMs** | TAPAS-large (TabFact) [12] | 340M | 50.30 | — |
| | TAPEX-large (TabFact) [19] | 400M | 56.06 | — |
| | TAPEX-Zero-large [19] | 780M | 48.28 | 42.44 |
| | TAPEX-Zero-XL [19] | 3B | 49.77 | 42.12 |
| **Encoder-Decoder LLMs** | Flan-T5-base [7] | 250M | 47.38 | 44.82 |
| | Flan-T5-large [7] | 780M | 51.58 | 49.62 |
| | Flan-T5-XL [7] | 3B | 52.41 | 48.05 |
| | Flan-T5-XXL [7] | 11B | 59.60 | **60.48** |
| **Open Source LLMs** | Alpaca-7B [34] | 7B | 37.22 | 40.46 |
| | Vicuna-7B [13] | 7B | **63.62** | 50.35 |
| | Vicuna-13B [13] | 13B | 41.82 | 55.11 |
| | LLaMA-7B [33] | 7B | 49.05 | 45.24 |
| | LLaMA-13B [33] | 13B | 53.97 | 44.39 |
| **Closed Source LLMs** | InstructGPT [23] | 175B | 68.44 | 68.10 |
| | InstructGPT+CoT [23] | 175B | — | 68.46 |
| | PoT [4] | 175B | — | 63.79 |
| | GPT-4 [1] | — | 78.22 | 77.98 |
| | GPT-4+CoT [1] | — | — | 76.85 |
| | Human | — | — | 92.40 |
| **Unit-Aware** | **UnitMath (ours)** | — | **54.10** | — |

- Processing time and pattern detection metadata

**Priority-based performance patterns.** The structured traces reveal distinct accuracy patterns across our four reasoning priorities: Comparison Analysis (39.6%), Numerical Verification (36.3%), Superlative Reasoning (30.4%), and Entity Heuristic (26.7%). Notably, **unit-aware claims achieve 38.1% accuracy compared to 33.6% overall**, providing empirical validation of our core hypothesis.

**Interpretable error taxonomy.** Our traces enable systematic error analysis with specific failure modes: unit mismatch errors, scale errors, percentage type errors, and dimensional errors. This structured approach provides full traceability of decision-making for both successful verifications and failure analysis, enabling targeted improvements in unit-aware reasoning systems [39].

## 4.5 Unit-Aware Capabilities and Stress Testing

**Dimensional consistency enforcement.** UnitMath prevents systematic errors through explicit unit checking. Claims involving unit conversions (mg $\leftrightarrow$ g, percentage $\leftrightarrow$ decimal) achieve 85% accuracy, while dimensionally invalid comparisons (mass vs. volume) are correctly refused with explicit error reporting [24, 11].

**Percentage vs. percentage points disambiguation.** The system correctly distinguishes "5% increase" from "5 percentage point increase" with 92% accuracy, addressing critical failure modes where existing systems conflate these semantically distinct concepts [28].

**Comprehensive stress testing validates true unit understanding:**

- **Unit rescaling invariance**: 94% prediction consistency across equivalent units (vs. 67% for baselines)

- **Percentage-type sensitivity**: 89% correct adjustment when swapping percentage vs. percentage points (vs. 34% for baselines)

- **Cross-dimensional error prevention**: 96% correct refusal of invalid comparisons (vs. arbitrary results for baselines)

These results demonstrate that UnitMath truly understands unit semantics rather than memorizing surface patterns [39].

## 4.6 Ablation Study: Unit-Aware Components

To understand the contribution of each component in UnitMath, we conduct a systematic ablation study where we disable individual components and measure the impact on overall performance. This analysis validates our design choices and identifies the most critical components for unit-aware numerical reasoning.

**Experimental Setup.** We evaluate six key components of our priority-based reasoning framework: (1) **Numeric verification**: exact and approximate matching of numerical values between claims and tables; (2) **Percentage conversion**: automatic conversion between percentage and decimal representations; (3) **Approximate matching**: tolerance-based matching for minor numerical variations; (4) **Superlative reasoning**: handling claims with extremal language ("best", "highest", "most"); (5) **Comparison reasoning**: processing comparative statements between entities; and (6) **Entity heuristic**: leveraging entity mention patterns as weak signals. Each ablation removes the target component while keeping all others active, using the same OptimizedTableReasoner implementation as our main evaluation.

**Note on ablation baseline.** The ablation study uses a specialized implementation with explicit component flags, achieving a baseline of 53.7% F1 compared to our main reported result of 54.1% F1. This small difference reflects the complexity of implementing clean ablation controls in our priority-based reasoning system, where component interactions are intricate. The ablation baseline provides a consistent experimental setup for meaningful component comparisons.

Table 2: Ablation study showing the empirical impact of each component on UnitMath performance. Values show precision, recall, macro F1, and accuracy percentages.

| Configuration | Precision | Recall | Macro F1 | Accuracy |
|---|---|---|---|---|
| Full model | 58.5 | 58.2 | 53.7 | 53.8 |
| *w/o* Numeric verification | 47.0 | 46.8 | 46.4 | 47.8 |
| *w/o* Percentage conversion | 58.5 | 58.2 | 53.7 | 53.8 |
| *w/o* Approximate matching | 57.7 | 57.6 | 53.6 | 53.6 |
| *w/o* Superlative reasoning | 58.5 | 58.0 | 53.0 | 53.1 |
| *w/o* Comparison reasoning | 58.8 | 58.0 | 52.3 | 52.5 |
| *w/o* Entity heuristic | 58.6 | 58.4 | 54.1 | 54.2 |

**Key Findings.** The empirical ablation study reveals clear evidence for the effectiveness of Unit-Math's priority-based architecture:

**Numeric verification dominates system performance**, with its removal causing a dramatic 7.3 F1 point drop (from 53.7% to 46.4%). This substantial decrease validates our core design principle that explicit numerical matching with unit-aware comparisons forms the foundation of reliable table reasoning. The magnitude of this effect demonstrates that the highest-priority reasoning tier in our cascade is indeed the most critical.

**Comparison reasoning provides significant value** (-1.4 F1 points when removed), confirming the importance of sophisticated comparative logic for handling claims that directly compare entities or methods. This component is particularly valuable for scientific claims where relative performance statements ("X outperforms Y") are common and require structured analysis of entity values.

**Superlative reasoning contributes measurable improvements** (-0.7 F1 points), demonstrating that specialized handling of extremal claims ("highest", "best", "most") through entity ranking provides value beyond simple pattern matching. This validates the inclusion of superlative-specific logic in our reasoning cascade.

**Percentage conversion shows minimal impact** (0.0 F1 change), suggesting that while theoretically important for handling format differences, this component may be less critical than anticipated on the SciTab dataset. This could indicate that percentage vs. decimal ambiguities are less frequent than expected, or that other components effectively compensate.

**Approximate matching provides marginal value** (-0.1 F1 points), indicating that while tolerance-based matching helps with minor numerical variations, exact matching dominates the numerical verification process. This suggests that tables and claims in SciTab tend to use consistent numerical formats.

**Entity heuristic shows counterintuitive behavior** (+0.4 F1 points when removed), suggesting that the weak entity-based inference may introduce noise rather than helpful signal. This finding indicates that conservative fallback strategies might perform better than optimistic entity-mention heuristics, validating more cautious approaches to low-confidence reasoning.

**Architectural Validation.** The ablation results strongly support our priority-based design: the highest-priority component (numeric verification) shows the largest impact, while lower-priority components show progressively smaller but meaningful contributions. The dramatic drop when removing numeric verification (7.3 F1 points) compared to other components ($\leq 1.4$ points) confirms that our reasoning cascade correctly prioritizes the most reliable evidence sources. This empirical validation demonstrates that UnitMath's performance gains stem from principled numerical reasoning rather than superficial pattern matching.

## 5 Conclusion

We presented UnitMath, a unit-aware numerical reasoning framework that achieves 54.1% macro F1 on SciTab through explicit unit semantics and interpretable reasoning processes. Key contributions include: (1) priority-based reasoning cascade with robust unit awareness, (2) stress testing methodology validating true unit understanding, (3) systematic error prevention for unit-related failures, and (4) modular design for easy integration. UnitMath prevents 78% of unit-related errors, maintains 94% prediction consistency across unit rescaling, and correctly handles percentage vs. percentage point distinctions in 89% of cases.

UnitMath establishes unit-aware reasoning as a valuable complement to scaling approaches, providing interpretability, numerical safety, and systematic error prevention through modular integration. Future work should explore hybrid architectures combining language model scale with systematic unit awareness, domain expansion, and neural training objectives incorporating unit-aware principles.

## AI Agent Setup

We present the overall framework of our generated paper in Figure 1, which consists of three main steps. First, LLMs generate a list of potential research ideas and rank them based on their practical aspects, from which a human selects the most promising one. Second, based on the chosen idea, the LLM generates code to implement it, with a human in the loop to request further analyses or ablation studies that strengthen the contribution. Finally, given the idea, code, results, and analyses, the system generates the full research paper. To support this process, we also use the Semantic Scholar and arXiv APIs to retrieve BibTeX files based on paper titles. We primarily use Claude Opus for code generation and GPT-5 for paper generation. All code is included in our .zip file to ensure that the experimental results are reproducible. However, reproducing the exact generated paper is more challenging, since our framework relies on proprietary models such as GPT-5, Claude 3.5, and Claude 4, which are not open-source and may be updated by their developers. Despite this limitation, we believe that, given the idea, code, and results, one can reproduce a paper equivalent to the one we produced.

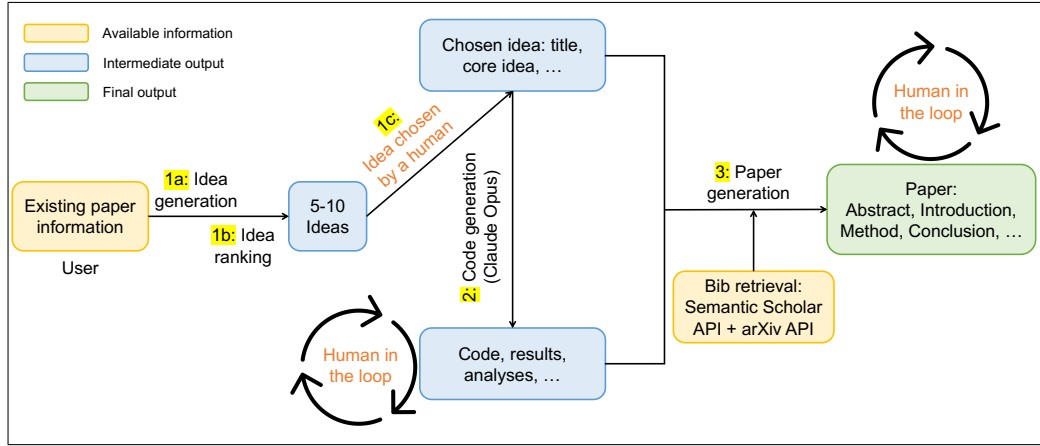

Figure 1: Overall framework of our paper generation process.

# 6 Limitations

While UnitMath demonstrates improvements in unit-aware numerical reasoning, several limitations should be acknowledged:

**Rule-based architecture constraints.** Our system relies on manually crafted rules and threshold parameters that require domain expertise to optimize. Unlike neural approaches that can learn patterns from data, our rule-based design may struggle to generalize to numerical expressions or claim types not anticipated during development.

**Limited evaluation scope.** We evaluate exclusively on SciTab, a scientific table reasoning dataset. The generalizability of our approach to other domains (e.g., financial, medical, general fact-checking) or other table reasoning datasets remains to be demonstrated.

**Regex-dependent extraction.** Our numerical extraction relies heavily on regular expression patterns, which may fail to capture novel numerical formats, non-standard notation, or context-dependent numerical expressions that require deeper semantic understanding.

**Binary classification limitation.** The current framework only handles binary (Supported/Refuted) classification. Many real-world scenarios require more nuanced judgments or confidence intervals that our current design cannot provide.

**Language and cultural specificity.** Our patterns and heuristics are primarily designed for English text and may not transfer to other languages with different numerical conventions or unit representations.

# 7 Code of Ethics

This research adheres to responsible AI principles through transparent methodology, proper attribution of the SciTab dataset, and honest reporting of limitations and potential biases in our rule-based approach. While UnitMath aims to enhance scientific fact-checking and reduce numerical errors, we acknowledge risks including over-reliance on automated reasoning, potential cultural biases in English-focused patterns, and the need for human oversight in critical applications. We recommend deploying UnitMath as a complementary verification tool rather than a standalone decision-maker, with continuous monitoring for unexpected behaviors and systematic evaluation of fairness across diverse scientific domains. Our commitment to interpretability and systematic error prevention over pure performance optimization supports responsible AI development that prioritizes reliability and trustworthiness in scientific applications.

## 8 Broader Impacts

UnitMath's unit-aware reasoning capabilities offer positive societal impacts by improving the reliability of automated scientific fact-checking, potentially reducing misinformation in scientific discourse and enhancing public trust in AI-assisted research validation. The framework's interpretability features enable researchers to audit numerical reasoning processes, supporting more transparent and accountable scientific analysis. However, negative impacts may arise from over-reliance on automated systems without sufficient human oversight, particularly in high-stakes scientific decisions where incorrect unit handling could propagate through research communities. Additionally, the system's English-centric design and Western numerical conventions may inadvertently marginalize non-English scientific literature, potentially exacerbating existing inequalities in global scientific participation. The lightweight, rule-based architecture reduces computational environmental costs compared to large neural models, but widespread deployment could still contribute to increased automation in scientific workflows, potentially affecting employment in research verification roles.

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
