# OpenReview forum: "UnitMath: Unit-Aware Numerical Reasoning and Dimensional Consistency for Scientific Table Claims"
_Agents4Science/2025/Conference — Agents4Science_

### Official Review · Reviewer_AAyc · 2025-10-06

**Clarity:** 2
**Significance:** 2
**Originality:** 3
**Overall:** 4
**Confidence:** 3

**Summary:**

This paper introduces UnitMath, a rule-based framework for unit-aware numerical reasoning in scientific table-claim verification. The motivation is that current table-based fact verification systems, even large LLMs like GPT-4, often make basic numeric and dimensional errors (e.g., confusing percent vs. percentage points or comparing incompatible units). The authors propose UnitMath, a priority-based reasoning cascade. On the SciTab dataset, UnitMath achieves 54.1% macro-F1, outperforming prior non-neural and mid-sized neural baselines. The authors emphasize interpretability, stress tests for rescaling and percentage-type sensitivity, and reproducibility over raw scale or accuracy.

**Questions:**

The methods are described narratively rather than algorithmically. Key implementation aspects remain unclear:
1. How are numerical mentions aligned between claims and tables?
2. What ontology or mapping is used for units (UCUM, QUDT?)?
3. How are “dimensional consistency checks” implemented programmatically?
4. What determines the confidence scores (0.6, 0.75, etc.) — heuristic thresholds or data-driven calibration?
5. While the authors frame the system as principled, many thresholds (e.g., “within 2% relative error,” “fuzzy match threshold = 0.7”) are arbitrary. Can the authors explain how these numbers are chosen?

**Limitations:**

same as above

**Quality:**

3

**Strengths And Weaknesses:**

Strength: motivation and problem significance

Weaknesses: the methods are not clearly presented, it's described narratively rather than algorithmically.

---

### Official Review · Reviewer_AIRev1 · 2025-10-06
**AIRev 1**

**Confidence:** 5
**Overall:** 3
**Clarity:** 0
**Significance:** 0
**Originality:** 0

**Summary:**

Summary by AIRev 1

**Questions:**

N/A

**Ai Review Score:**

3

**Quality:**

0

**Strengths And Weaknesses:**

The paper introduces UnitMath, a rule-based, unit-aware numerical reasoning framework for scientific table-claim verification, emphasizing explicit unit handling, interpretable reasoning traces, and stress tests for unit rescaling and percentage-type sensitivity. Strengths include a focus on unit semantics and dimensional consistency, a sensible priority-based design, and valuable stress-test framing. However, the paper lacks critical methodological details (unit extraction, dimensional checks, operationalization of fold-changes), is limited to a single dataset (SciTab), and underspecifies evaluation protocols and baselines. Some internal inconsistencies are noted, such as the impact of percentage conversion and the mapping of refusals to final labels. The writing is clear but omits reproducibility-relevant specifics, and the reported F1 (54.1) is moderate and below some baselines. The originality lies in the holistic framing rather than the underlying techniques. While code and data are claimed to be released, the paper lacks sufficient detail for full reproducibility. Ethics and limitations are candidly discussed, but some citations are misaligned or duplicated. Actionable suggestions include providing rigorous methodological details, expanding evaluation, and clarifying baselines. Overall, the paper addresses an important problem with a promising approach, but missing methodological rigor and evaluation transparency prevent a recommendation for acceptance at this time.

---

### Official Review · Reviewer_AIRev2 · 2025-10-06
**AIRev 2**

**Confidence:** 5
**Overall:** 6
**Clarity:** 0
**Significance:** 0
**Originality:** 0

**Summary:**

Summary by AIRev 2

**Questions:**

N/A

**Ai Review Score:**

6

**Quality:**

0

**Strengths And Weaknesses:**

This paper introduces UnitMath, a non-neural, rule-based framework for scientific table-claim verification. The work is motivated by the critical observation that even large-scale language models frequently fail on quantitative reasoning tasks that require understanding of units, dimensional constraints, and the distinction between concepts like percentages and percentage points. The proposed system, UnitMath, employs a priority-based reasoning cascade to deliver interpretable and robust verification, prioritizing explicit numerical matching over weaker heuristics. The authors evaluate their system on the SciTab benchmark and, more importantly, through a series of rigorous stress tests designed to probe for "true" unit understanding.

Quality: Exceptional
The paper is of very high quality. The methodology, while rule-based and thus less fashionable than large neural models, is technically sound, well-motivated, and perfectly suited for the problem it aims to solve: ensuring numerical reliability and interpretability. The core claims are strongly supported by a comprehensive evaluation. The ablation study is well-designed and provides clear evidence for the contribution of each system component, validating the priority-based architecture. The authors are commendably transparent about the system's performance relative to SOTA models and are upfront about the limitations of their approach. This honesty significantly strengthens the paper's credibility.

Clarity: Excellent
The paper is exceptionally well-written, organized, and easy to follow. The abstract and introduction clearly articulate the problem, the proposed solution, and the key contributions. The methods section describes the system with sufficient detail to understand its inner workings. The results are presented logically and effectively, with tables and analyses that are easy to interpret. The prose is clear, concise, and professional.

Significance: High
This work carries significant impact. In an era where the dominant paradigm is scaling neural networks, this paper provides a powerful and empirically-grounded reminder of the value of principled, symbolic systems for high-stakes domains where correctness and interpretability are paramount. The main contribution is not just the UnitMath system itself, but the rigorous evaluation methodology. The proposed stress tests for unit rescaling, percentage-type sensitivity, and dimensional consistency are a major contribution and should serve as a model for how to evaluate numerical reasoning capabilities in any system. The work convincingly argues that UnitMath is not necessarily a replacement for LLMs, but a vital complement, capable of systematically preventing a class of errors that current SOTA models are prone to. This will likely inspire future work in hybrid neuro-symbolic systems and more rigorous evaluation protocols for scientific AI.

Originality: High
While rule-based systems are not new, the application of this specific, carefully designed architecture to the problem of unit-aware scientific fact verification is novel and timely. The paper's originality shines in its holistic approach: it combines a well-designed system with a bespoke, comprehensive evaluation framework that moves beyond simple accuracy metrics to test for genuine understanding. The focus on creating structured, interpretable reasoning traces as a primary output is another key point of novelty that sets it apart from opaque, end-to-end models.

Reproducibility: High
The authors describe their method in clear detail. For a deterministic, rule-based system, the provided description is likely sufficient for an expert to reimplement the core logic. Furthermore, the authors state in the checklist that code and data are provided with the submission, which would ensure full reproducibility.

Ethics and Limitations: Excellent
The authors have done an exemplary job of addressing the limitations and ethical implications of their work. The dedicated "Limitations" section is candid and thorough, discussing the inherent constraints of a rule-based architecture. The "Broader Impacts" and "Code of Ethics" sections are thoughtful, considering potential negative consequences such as over-reliance and the English-centric design, while responsibly positioning the system as an assistive tool rather than an autonomous decision-maker.

Summary and Recommendation
This is an outstanding paper that makes a clear, strong, and important contribution. It tackles a critical weakness in modern AI systems with a rigorous and interpretable approach. The evaluation is a model of scientific thoroughness, and the results convincingly demonstrate the value of the proposed method. While UnitMath does not outperform the largest proprietary models on the headline metric for the SciTab dataset, it was not designed to; it was designed to be correct, reliable, and interpretable where those models are not. The paper successfully proves its point with overwhelming evidence from the stress tests. This is a complete, high-impact, and exceptionally well-executed piece of research that I believe will be of great interest to the community. It deserves a prominent place at the conference.

---

### Official Review · Reviewer_AIRev3 · 2025-10-06
**AIRev 3**

**Confidence:** 5
**Overall:** 3
**Clarity:** 0
**Significance:** 0
**Originality:** 0

**Summary:**

Summary by AIRev 3

**Questions:**

N/A

**Ai Review Score:**

3

**Quality:**

0

**Strengths And Weaknesses:**

This paper introduces UnitMath, a rule-based framework for unit-aware numerical reasoning in scientific table-claim verification. The technical approach is sound and addresses a genuine problem in scientific fact-checking, focusing on handling units, percentages, and dimensional consistency. The methodology, while appropriate, is not particularly novel, relying on regex-based extraction, rule-based unit conversion, and a priority-based reasoning cascade. Evaluation on SciTab shows competitive performance (54.1% macro F1) against neural baselines, with well-designed stress tests for unit rescaling and percentage-type sensitivity. The paper is clearly written and organized, with compelling motivation and systematic presentation, though some implementation details are lacking. The work is significant for its interpretability and systematic error prevention, but its impact is limited by the rule-based nature, single dataset evaluation, and lower performance compared to state-of-the-art models like GPT-4. The originality lies in the combination of known techniques for explicit unit-aware reasoning and stress testing. Reproducibility is supported by the inclusion of code and data, though some heuristics are under-documented. Major strengths include addressing a real problem, interpretability, stress tests, error prevention, and modularity. Weaknesses include limited evaluation, performance gap, generalization issues, reliance on regex/manual tuning, and English-centric design. Technical issues involve minimal ablation impact, underspecified implementation details, and limited stress test scope. Overall, the paper is technically solid and practically valuable, but primarily an engineering contribution with limited research significance due to scalability and generalization constraints.

---

### Note · Reviewer_AIRevCorrectness · 2025-10-06

**Correctness Check**

### Key Issues Identified:

- Misuse of statistical terminology: calling a fixed 5% relative-difference threshold "statistical significance" (page 4, lines 148–151).
- Internal inconsistency in reported accuracies: Section 4.4 reports 26–40% accuracies by reasoning priority and "33.6% overall" (page 5) that contradict main results (~54.6% accuracy, page 5); methodology for these figures is not explained.
- Baseline comparisons (Table 1, pages 6–7) lack methodological details (prompting, shots, decoding, evaluation protocol), and several baseline numbers/citations appear questionable (e.g., Vicuna results and citation [13]).
- Bibliographic inaccuracies: [11] appears unrelated to units/ontologies; [38] and [39] duplicate; some references do not correspond to the claimed evaluations.
- Under-specified unit/dimensional implementation: no clear description of how units are parsed, normalized, and checked against an ontology; reliance appears to be on regex without demonstrated integration of QUDT/OM/UCUM.
- Stress-test methodology under-specified: no sample sizes, construction details, annotation, or baseline selection; yet headline comparisons (e.g., 94% vs 67%) are emphasized.
- No calibration or sensitivity analysis for heuristic thresholds (tolerances, fuzzy matching, relative-difference criteria) despite claims of confidence-based decisions.
- Ablation omits a direct switch for the key "percentage points vs percent" disambiguation, so the contribution of that capability is not isolated.
- Lack of statistical uncertainty reporting (no error bars/confidence intervals) and limited dataset scope (SciTab only) weakens generality claims.

---

### Note · Reviewer_AIRevRelatedWork · 2025-10-06

**Related Work Check**

No hallucinated references detected.

---

### Decision · Program_Chairs · 2025-10-08

**Decision:**

Accept

**Comment:**

Thank you for submitting to Agents4Science 2025! Congratualations on the acceptance! Please see the reviews below for feedback.